# Treatment of Richter Transformation of Chronic Lymphocytic Leukemia in the Modern Era

**DOI:** 10.3390/cancers15061857

**Published:** 2023-03-20

**Authors:** Robert Briski, Justin Taylor

**Affiliations:** 1M.D. Anderson Cancer Center, Houston, TX 77030, USA; 2Sylvester Comprehensive Cancer Center, University of Miami Miller School of Medicine, Miami, FL 33136, USA

**Keywords:** targeted therapy, Richter transformation, precision medicine, chronic lymphocytic leukemia, Bruton’s tyrosine kinase inhibitors, checkpoint inhibitors, bispecific antibodies, stem cell transplant, chimeric antigen receptor t-cells

## Abstract

**Simple Summary:**

Though rare, Richter Transformation (RT) is a serious complication of chronic lymphocytic leukemia. RT carries a high mortality rate and represents an unmet clinical need. With an improved understanding of its molecular biology, there is hope that novel targeted therapies and immunotherapies will lead to improved outcomes in this disease. This review outlines what is known about RT as well as where the field is headed. An illustrative case is also included to facilitate the reader’s understanding of the material presented.

**Abstract:**

Richter Transformation (RT) refers to the development of an aggressive lymphoma in the setting of chronic lymphocytic leukemia (CLL). While many variants of RT are recognized, diffuse large B-cell lymphoma (RT-DLBCL) is the most common (80%), followed by Hodgkin’s lymphoma (RT-HL, 19%). Diagnosis is based upon histologic evaluation of clinically suspicious lymph nodes. Positron emission tomography (PET) may be used to select the node of interest for biopsy. Although clonality testing is not a prerequisite of RT diagnosis, it has significant implications for survival. Clonally related DLBCL carries the worst prognosis with a median overall survival (OS) of less than one year in the era of combination chemotherapies with or without anti-CD20 antibodies. Prognosis has improved with the use of stem cell transplant and newer agents such as targeted therapy and newer forms of immunotherapy. Consideration of a clinical trial is encouraged. This review describes our current understanding of RT and focuses on treatment of RT-DLBCL, including clinical trials in progress and new therapies in development. We also report an illustrative example of a patient with clonally related DLBCL who survived two years after diagnosis without the use of combination chemotherapy.

## 1. Clinical Case

A 65-year-old gentleman with previously treated non-Hodgkin’s lymphoma (unknown subtype, felt to be in complete remission for the last 35 years) presented to an outside hospital with B symptoms (night sweats and unintended weight loss ≥10% of body weight in 6 months) and axillary lymphadenopathy. A complete blood count was performed, which demonstrated a lymphocytosis of 14,190, a normal platelet count and a normal hemoglobin, suspicious for chronic lymphocytic leukemia/small lymphocytic lymphoma (CLL/SLL). The axillary lymph node was biopsied and was positive for kappa restriction, CD5, CD19, CD20, CD23 and CD11c and negative for CD10 by flow cytometry (FC), confirming the diagnosis of CLL/SLL. Ancillary testing for prognostication purposes demonstrated a 17p deletion by fluorescent in-situ hybridization (FISH), an unmutated immunoglobulin heavy chain variable region (IGHV), and TP53 and SF3B1 mutations by next generation sequencing. The NOTCH1 gene was unmutated, and a karyotype was not available. This pattern was consistent with biologically more aggressive CLL/SLL, suggestive of the probable need for future treatment. However, given his clinically indolent RAI stage I disease, he was observed. After four months of observation, he developed abdominal pain, which prompted a CT scan of his abdomen and pelvis. This demonstrated a conglomerate mesenteric nodal mass measuring 11.6 cm × 5.8 cm. Unfortunately, a PET/CT was not performed. The patient was started on ibrutinib monotherapy. He remained on treatment for three months until recurrent abdominal symptoms led to a repeat CT demonstrating progression of his mesenteric lymphadenopathy. A CT-guided core needle biopsy of the mesenteric lymph node was consistent with histologically accelerated CLL (HAC). The patient was then referred to our institution.

Upon presentation, he was with B symptoms, a hemoglobin of 11.5, platelets of 177, a white blood cell count of 16.3 and an LDH of 409. Re-review of the outside biopsy confirmed the diagnosis of HAC, and the decision was made to start the patient on venetoclax monotherapy. After four months of treatment, his blood counts and LDH normalized. He underwent a restaging PET/CT which demonstrated reduction in the size of the lymph nodes (including the largest node by ≥50%) with an SUV_max_ of 4.4. A bone marrow (BM) biopsy was performed and demonstrated a low level of involvement of CLL by FC (0.048% of the WBC in the marrow). Interestingly, the 17p deletion was no longer present. Overall, this was consistent with a partial remission. He then underwent consolidation therapy with a haploidentical allogeneic stem cell transplant using fludarabine, cyclophosphamide and total body irradiation as the conditioning regimen. Response assessment with PET/CT and BM biopsy was negative for measurable residual disease at day 30. Unfortunately, at day 100, a repeat BM biopsy demonstrated relapsed kappa restricted CLL.

The patient was subsequently started on retreatment with venetoclax for CLL, however, after three months of therapy, he developed back pain and was found to have progressive retroperitoneal lymphadenopathy on PET/CT with a SUV_max_ of 9.1. A core needle biopsy was obtained and consistent with kappa restricted diffuse large B-cell lymphoma (DLBCL). IGHV gene rearrangements were compared with a prior CLL/SLL sample, and clonally related DLBCL variant Richter Transformation (RT) was confirmed.

## 2. Introduction

Richter transformation (RT) is defined as an aggressive lymphoma in the setting of chronic lymphocytic leukemia (CLL). Currently, there is no requirement to demonstrate a clonal relationship between an aggressive lymphoma and the underlying CLL. Thus, all cases, whether they represent two separate primary malignancies or a true transformation, are grouped as RT. The prognosis of RT is particularly poor when CLL transforms into a clonally related diffuse large B-cell lymphoma (RT-DLBCL). Fortunately, RT is uncommon, occurring in 2–10% of CLL patients [1,2,3,4,5,6,7]. This review focuses on the evolution of RT diagnosis, prognostication and treatment in the modern era.

## 3. Historical Background of Richter Transformation

In 1928, Dr. Maurice Richter published a sentinel case report of a 46-year-old gentleman afflicted by lymphocytic leukemia and rapidly progressing lymphadenopathy [8]. The lymphoproliferative disease was so aggressive it resulted in the patient’s death one month after presentation. A postmortem autopsy demonstrated diffuse peripheral and mesenteric lymphadenopathy (with bulky lymph nodes up to 8 cm in size) as well as hepatosplenomegaly. Microscopic evaluation of the lymph nodes revealed effacement of nodal architecture with several discrete small lymphocytes admixed large irregular cells of undetermined origin. Dr. Richter incorrectly suggested these large irregular cells were derived from “reticular cells” (which he described as connective tissue) and classified it as “reticular sarcoma.” He postulated the large irregular cells emerged after the patient’s leukemia, however, was unable to determine their relation to the leukemia. 

Over the next 40 years, several clinicians would go on to describe similar findings in their patients without a clear connection made between these two disease processes. Finally, in 1964, Dr. Lortholary published a case series of four patients; some of who transformed into Hodgkin’s lymphoma, and some of who transformed into what he described as reticular sarcoma (though if one were to review the pathology images published in his paper, at least one case looks like a DLBCL) [9]. Lorthoraly suggested this transformation was related to the underlying lymphocytic leukemia and referred to the transformation as Richter’s Syndrome (RS). From then on, the liberal definition of any aggressive lymphoma in the setting of chronic lymphocytic leukemia was considered RS. The majority of cases reported in literature are diffuse large B-cell lymphoma (80–90%) [10]. The next most common are Hodgkin’s lymphoma (10–20%), followed by rare case reports of Burkitt’s lymphoma and other aggressive lymphomas such as T-cell lymphoma (collectively ≈1% of cases) [11,12,13,14,15,16]. In 2007, it was finally demonstrated that some, but not all, cases of RS were clonally related to the underlying CLL [17]. Since this disease is felt to represent a transformation from one neoplasm to another, the WHO recommends the term Richter Transformation (RT) [18]. Though clonality is not a diagnostic requirement of RT, many contend that clonally unrelated disease is not a true transformation as it behaves more like de novo disease and carries a better prognosis. (This will be discussed further in the prognosis section) [19]. Because the majority of cases of CLL transformation are to DLBCL, we will focus this review on RT to DLBCL, unless specifically mentioned otherwise.

## 4. Predictability of Richter Transformation at CLL Diagnosis

The ability to predict the risk of RT at CLL diagnosis would be important for two reasons. First, it would allow an investigator to accurately assess a therapy’s ability to delay RT based on the patient’s risk of RT. Second, if a treatment was proposed to prevent RT, it could be given based upon the patient’s risk. Unfortunately, there are no standardized models to predict the risk of RT at CLL diagnosis, with 5-year estimates ranging from 13–78.9% for high risk groups [6,20,21]. To date, there are five major clinical studies that investigate the risk of RT in treatment-naive CLL patients. Two of these studies did not adjust for confounding variables with a multivariate analysis; thus, they will not be discussed [1,22]. The other three have slightly divergent results as a result of study design and will be described herein.

Collectively, these three studies identified the following six risk factors for RT: lymph node size ≥3 cm, various abnormal CLL FISH patterns (see below), mutational status and stereotypic pattern of the immunoglobulin heavy chain variable region (IGHV), TP53 disruptions (mutations and or deletions), NOTCH1 mutations and complex karyotype (CK, defined as ≥3 chromosomal abnormalities within the same clone) (Table 1). Unfortunately, there is no single multivariate analysis to assess these variables together.

The only study to include lymph node size (≥3 cm) demonstrated lymph node size and the lack of a del(13q) to be statistically significant risk factors for RT in a multivariate analysis. They failed to demonstrate statistical significance for TP53 disruptions, IGHV mutational status, and IGHV stereotype pattern. NOTCH1 mutations and CK were not included in their study. The estimated 5-year risk of RT for patients with lymph nodes ≥3 cm and a concomitant cytogenetic profile which lacks a del(13q) was 78.9%. However, the authors conclude that the small sample size and lack of follow-up limits the validity of this model [6].

The only study to include NOTCH1 and SF3B1 mutations, handled their CLL FISH profile differently from the other two studies. Rather than assess each cytogenetic abnormality independently, they combined TP53 disruptions with trisomy 12 and 11q deletions as a composite abnormal CLL FISH profile. They found NOTCH1 mutation, the IGHV stereotype 4–39 and the composite abnormal FISH profile to be adverse risk factors by multivariate analysis. NOTCH1-mutated CLL patients had an estimated 5 year RT risk of 18.6%. Patients with both a NOTCH1 mutation and a IGHV4-39 stereotype were projected to have an even higher risk, but the sample size was too small to make reliable estimates. Lymph node size and CK were not included in this study [20].

Only one study evaluated CK and found CK, TP53 disruptions, Binet stage B-C, unmutated IGHV and del(11q) to be adverse risk factors by multivariate analysis. This study did not include lymph node size, IGHV stereotypic pattern nor NOTCH1 mutations. They generated a scoring system, in which patients with CK had the highest risk score with an estimated 5-year risk of RT of 13% [21]. It is worth noting that these models are more Kaplan-Meier statistical estimates than true reality due to the small sample sizes and lack of follow-up for intracohort subgroups. Further studies are needed to construct a more agreed upon model: Such a study should include all relevant clinical and molecular variables mentioned above.

## 5. Diagnostic Evaluation

RT should be considered when CLL patients present with rapidly proliferating lymphadenopathy (>3 cm), particularly when one nodal region is more afflicted than others, and one or more of the following are present: systemic symptoms, elevated LDH or hypercalcemia. If RT is suspected, a PET/CT is ordered as a rule out test based on the maximum standardized uptake (SUV_max_) to decide if a biopsy is warranted (with a negative predictive value (NPV) upwards of 90%) [23,24,25]. The appropriate SUV_max_ threshold for biopsy has been a subject of debate and seems to be patient-specific. The historic approach utilized a universal SUV_max_ cutoff of ≥5 to pursue a biopsy with a positive predictive value (PPV) of 38–53% and a NPV of 92–97% [23]. However, a subsequent study by Falchi et al. sparked a debate for changing the SUV_max_ cutoff to ≥10. In their study of patients with histologically indolent CLL (HIC), histologically accelerated CLL (HAC, described below) and RT, the median SUV_maxs_ were 3.7, 6.8 and 17.6, respectively. In turn, many CLL patients with a SUV_max_ of 5 underwent unnecessary biopsies demonstrating HAC opposed to RT [24]. Michallet et al. validated these results and demonstrated an improved PPV of 60% with a NPV of 99% when a SUV_max_ of 10 was used [25]. However, this was prior to treatment with kinase inhibitors (KI) such as Bruton’s tyrosine kinase inhibitors or phosphoinositide 3-kinase inhibitors (PI3Ki). Mato et al. demonstrated the NPV drops from 99% to 50% post KI treatment [26]. Thus, many centers will reserve biopsies for nodes with an SUV_max_ ≥10 in untreated patients and ≥5 in treated patients (particularly those treated with KIs). Of note, if a biopsy threshold is not met but the node remains suspicious, it is appropriate to follow the SUV_max_ during treatment with repeat scans. If the SUV_max_ in one or more nodes remains elevated while others decrease, it is appropriate to consider a biopsy.

Since RT is often focal, it is important to biopsy the most accessible lymph node with an SUV_max_ above the cutoff. Fine needle aspirates should not be used to obtain biopsies as they were found to be diagnostically inadequate in 53% of patients with RT [24]. At minimum, a core needle biopsy should be obtained, and consideration should be given for an excisional biopsy.

Histologic evaluation of RT-DLBCL classically reveals sheets of large cells (about 2–3 times the size of normal lymphocytes) with nuclei equal to or exceeding the size of a macrophage nucleus. These cells commonly resemble the proliferating cells of germinal centers (such as centroblasts), or less commonly immunoblasts. The centroblastic type typically have two to three prominent peripheral nucleoli and a narrow rim of basophilic cytoplasm, whereas the immunoblastic type tends to have a central nucleolus and abundant basophilic cytoplasm [27]. When these large cells are present in high numbers and distort nodal architecture, the diagnosis of DLBCL is quite clear. Less certain however, is when these cells appear spread out admixed large populations small lymphocytes which may distort the nodal architecture. In such circumstances, practitioners often question whether a DLBCL was missed as a result of sampling bias (i.e., either a core-needle biopsy or an excisional biopsy of one node in a nodal conglomerate was obtained). These “grey zone” cases have been described since 1988 and have more recently been referred to as histologically accelerated CLL (HAC) [28,29,30]. HAC is not recognized by the WHO classification and lacks formal diagnostic criteria. To provide more clarity into the diagnostic dilemma of HAC, Giné et al. attempted to define its morphology as well as to characterize clinical outcomes of HAC when compared with RT-DLBCL and histologically indolent CLL (HIC). They defined HAC as the presence of proliferation centers (PC) broader than a 20× field (or 0.95 mm^2^) and a high proliferation rate (Ki-67 > 40 percent or >2.4 mitoses/PC), as determined by Kaplan-Meier OS estimates. They were able to demonstrate that the OS for patients with HAC fell somewhere between that of patients with RT-DLBCL and patients without HAC, suggesting this disease entity stands alone as a separate process from RT-DLBCL. The difference in OS between HIC and HAC existed in the era of chemoimmunotherapy; however, it is unclear how things will change moving forward with new targeted agents. Unfortunately, Giné et al. were not able to determine whether patients with HAC have an increased risk of RT when compared to HIC.

## 6. Molecular Differences between Clonally Related RT-DLBCL and De Novo DLBCL

Though the histologic appearances of de novo DLBCL and RT-DLBCL are similar, there are some important molecular differences. These differences may help explain the more aggressive nature of RT-DLBCL, and more importantly, may offer actionable targets for RT-specific therapy. For instance, the expression of programmed cell death receptor (PD-1) on the surface of clonally related RT-DLBCL cells (seen in about 80% of cases) is typically absent on the surface of de novo DLBCL (seen in only 4% of cases) and represents an actionable target for RT-DLBCL specific therapy (see the therapeutics section for a more detailed description of therapies targeting PD-1). Mutations in TP53, NOTCH1, MYC and deletions in CDKN2A (cyclin-dependent kinase inhibitor 2A), as well as the preferential use of the 8th subset of stereotyped immunoglobulin heavy chains (IGHV4-39/IGHD6-13/IGHJ5), also seem to be more prevalent in clonally related RT-DLBCL [7,19,31,32,33]. Disruptions in TP53 occur in about 60–80% of clonally related RT-DLBCL cases [19]. TP53 is an important tumor suppressor gene, and its inactivation is felt to enable cell survival. This may partially explain the chemoresistance of RT-DLBCL. Some are hopeful that treatment with BCL-2 inhibitors will lead to TP53 independent cell death [34] and may help overcome some of this resistance (see the therapeutics section for a more detailed description of the BCL-2 inhibitor, venetoclax). The proliferative nature of RT-DLBCL may in part be explained by deletions in CDKN2A (seen in 20% of RT-DLBCL cases) [31], which allows for progression through the cell cycle and uncontrolled replication of neoplastic cells. This has led investigators to contemplate the use of cyclin-dependent kinase inhibitors for RT-DLBCL. Additionally, the proliferative nature of RT-DLBCL could also be due to disruptions in the MYC network, with MYC mutations occurring in 30% of cases of clonally related RT-DLBCL [19] and NOTCH1 mutations occurring in 30% of cases of clonally related RT-DLBCL [32]. Finally, the more prevalent use of the stereotyped eighth subset of heavy chain immunoglobulins in clonally related RT-DLBCL may suggest this conformation of the B-cell receptor promotes RT-DLBCL formation and survival [7]. It remains unknown how to target these last three differences. However, the molecular underpinnings of RT-DLBCL are still under investigation, and there is hope that new discoveries will lead to improved targeted therapeutic offerings.

## 7. Prognosis of Richter Transformation

One important prognostic factor is whether patients have received treatment for CLL prior to their RT. Treatment-naive patients experienced a superior overall survival (OS) compared with treated patients (46.3 months versus 7.8 months) [35]. Another important prognostic factor for OS is the clonal relationship of the DLBCL to the underlying CLL. In a retrospective analysis of 63 RT-DLBCL patients, those with clonally unrelated disease (n = 13) experienced a median OS of 65 months (similar to that of de novo DLBCL) versus a mere 14 months for those with clonally related disease (n = 50, *p*-value = 0.017) [19].

Given the importance of clonality to prognostication, we will describe techniques used to evaluate clonality. The traditional method utilized sequential testing, starting with flow cytometry to evaluate for divergent light chain expression (κ or λ) between a DLBCL and the underlying CLL. Divergent light chain restriction was felt to represent clonally unrelated disease. Conversely, if light chain restriction was found to be identical, a subsequent PCR analysis of the V–D–J segment of the heavy chain immunoglobulin was conducted to confirm disease clonality (in clonally related disease, V–D–J segments will be nearly identical). However, there was a case report which demonstrated the possibility of clonally related DLBCL and CLL having divergent light chain restrictions, suggesting that all cases should be evaluated more carefully regardless of light chain analysis [36]. Thus, it is now recommended that all cases be assessed by V–D–J rearrangement testing. The two techniques utilized for V–D–J segment analysis are real time PCR and next generation sequencing. To accurately perform this test, one must clearly demonstrate that the V–D–J segment being sequenced comes from an isolated DLBCL population and an isolated CLL population. This may not always be a trivial task. If the sample has an admixed population of DLBCL and CLL, one must clearly separate the cell populations prior to testing.

Because some centers lack technology for V–D–J segment sequencing, less ideal surrogates of disease severity have been proposed. He et al. proposed using program cell death-1 receptor (PD-1) expression as a proxy for clonality [37]. Interestingly, PD-1, which is usually found on the surface of T-cells, was found to be expressed on the surface of neoplastic B-cells in 80% of cases (12/15) of clonally related DLBCL, opposed to a mere 4% of non-RT-DLBCL cases (1/26). PD-1 expression on the surface of neoplastic cells as a proxy for V–D–J clonality testing has not been validated. Alternatively, a prognostic model for disease severity was proposed by Tsimberidou et al. [3]. This model stratifies patients based on five clinical/laboratory variables: Eastern Cooperative Group performance status > 1; number of CLL therapies ≥2; tumor size > 5 cm; serum LDH ≥1.5× the upper limit of normal; and platelets <100 × 10^9^/L. Each variable was weighted the same, and patients were given a score of 0-5, with median OS estimates of 1.1 years for low risk patients (score 0–1), 0.9 years for low-intermediate risk patients (score 2), 0.3 years for high-intermediate risk patients (score 3) and 0.1 years for high risk patients (score 4–5). While this scoring system was validated with a study at Mayo Clinic (n = 120) [38], its utility is a little more theoretical and research focused. Based on the dismal prognosis for low risk patients (median OS 1.1 years), most patients likely received treatment for their CLL and were probably with clonally related disease. It is unclear how useful this prognostic model would be for treatment-naive patients in which the clonality status is unknown. Finally, TP53 disruption has also been cited as a possible negative prognostic factor; however, more investigation with larger sample sizes is needed [19].

## 8. Historical Outcomes with Chemotherapy and Chemoimmunotherapy

There are no standardized phase III clinical trials to evaluate RT-specific treatments. Instead, data is derived from retrospective analyses and single-arm phase I/II prospective studies. This is due to the paucity of RT cases and a rapidly expanding number of treatments that could potentially be evaluated. Historically, RT was treated similar to other aggressive B-cell lymphomas, with several anthracycline-based, platinum-based and fludarabine-based regimens proposed [39,40,41,42,43,44,45,46,47]. Since these trials were not head-to-head comparisons, the only evaluations available are cross-study comparisons. Though such a technique poses its limitations (with respect to confounding variables such as study design and patient population), in general, outcomes were similar across chemotherapy and chemoimmunotherapy regimens (with more toxicities for non-R-CHOP regimens), leaving much room for improvement of RT treatments (Table 2).

A nice characterization of chemotherapy and chemoimmunotherapy outcomes can be illustrated by three studies published from the group at the MD Anderson Cancer Center (MDACC). These three studies include: a prospective study investigating the combination of fractionated cyclophosphamide, vincristine, liposomal daunorubicin and dexamethasone (HCVXD); a prospective study investigating the use of Rituximab in combination with HCVAD alternating with methotrexate and cytarabine (R-HCVAD); and one retrospective study evaluating chemotherapy vs chemoimmunotherapy in general. When comparing HCVXD (n = 29) to R-HCVAD (n = 30), there was a slight difference in complete remissions (CR) (38% vs. 27% respectively) but no major overall difference between OS (10 months versus 9 months) [41]. Seven years following these trials, they compared more general outcomes of chemotherapy (n = 79) to chemoimmunotherapy (n = 47) and, again, found no difference in terms of response or survival [3]. Even with the addition of the new type II anti-CD20 antibody obinutuzumab, there has been no change in outcome for chemoimmunotherapy [40]. Thus, future studies moved towards studying targeted therapies or immunotherapies for RT.

## 9. The Role of Stem Cell Transplant (SCT) in RT

As discussed above, the prognosis for patients with RT receiving combination chemotherapy +/− an anti-CD20 antibody is typically a median OS less than 1yr. However, in 2006, Tsimberidou et al. were able to demonstrate a significantly improved median OS of ≥3 years for select RT patients who underwent allogeneic SCT (alloSCT). Of 148 RT patients evaluated in their retrospective analysis, 17 underwent an alloSCT. These 17 patients were split into two groups: those who underwent SCT after achieving a CR/PR (n = 7) and those who underwent SCT as a salvage therapy (n = 10). The 3-year OS for those undergoing alloSCT in CR/PR was 75%. Whereas those who underwent alloSCT as salvage had a 3-year OS of 21%. Still, those who underwent alloSCT as salvage had an impressive outcome compared with the prior standard, as they were with a similar 3-year OS as those who achieved a CR/PR without a subsequent alloSCT (3-year OS of 27%) [3].

There have since been four retrospective studies to address the role of SCT in RT [48,49,50,51]. Three of these studies enabled a comparison of alloSCT as consolidative therapy (in CR or PR) versus salvage [48,49,50]. All three studies validated the hypothesis that alloSCT as a consolidative therapy confers a superior outcome. In fact, Herrera et al. were able to demonstrate incremental improvements in PFS and OS depending on the degree of remission at alloSCT (3 year PFS/OS for CR versus PR versus SD/Prog was 66%/77% versus 43%/57% versus 5/15% respectively, *p* < 0.0001) [49]. While two of the three studies failed to demonstrate a difference in OS between CR versus PR [48], it is worth noting that all studies are at risk of confounding due to the lack of clonality testing between the DLBCL and the underlying CLL. Further, some studies did not include mutational analyses or a karyotype analysis. Though these data may suggest the depth of remission at alloSCT impacts the outcome, MRD status at the time of alloSCT remains undefined (i.e., is a negative PET scan sufficient?).

Though there are two studies which evaluated outcomes for both alloSCT and autologous SCT (autoSCT) and reported similar outcomes for each approach, comparisons between alloSCT and autoSCT could not be made due to selection bias: patients with worse prognostic factors and clinically more aggressive disease were more commonly selected to undergo alloSCT [48]. In general, alloSCT has been the preferred SCT technique in RT given the percieved graft versus leukemia benefit (whereby the donor’s immune system will attack any residual host cancer cells). It is important to note that some patients may lack a good donor for an alloSCT, or there may be other clinical reasons in which an alloSCT may not be the best approach for a patient. In such circumstances, an autoSCT may be considered.

While SCT has improved outcomes for patients with RT-DLBCL, this approach is limited to fit/young patients (typically < 65 years of age), which unfortunately tends to represent less than one-half of the patient population.

## 10. Bruton’s Tyrosine Kinase Inhibitors (BTKis) 

Bruton’s tyrosine kinase plays a pivotal role in CLL cell survival via the B-cell receptor (BCR) signaling pathway (Figure 1). Its role is integral to the survival of CLL cells and inhibition with the first generation BTKi, ibrutinib, demonstrating an impressive ORR of 89% in patients with relapsed/refractory CLL [52]. Two major side effects of ibrutinib are bleeding and atrial fibrillation, which is the result of non-selective inhibition of tyrosine kinases found in platelets and myocardial tissue [53,54,55]. Efforts made to discover more selective inhibitors of the BTK have led to the discovery of newer generations of BTKis with improved side effect profiles, including acalabrutinib [55], zanabrutinib [56], and more recently pirtobrutinib [57,58]. While the use of BTKis in CLL has not led to a reduction in the incidence of RT [59], patients are still able to experience a response to BTK inhibition after transformation. The degree of response is less for those who received a BTKi pre-transformation [60].

The first published study investigating the role of BTKis in RT-DLBCL was a case series of 4 patients, the majority of whom had relapsed/refractory RT. Only 3 out of 4 patients were evaluable. All 3 evaluable patients responded (1 CR and 2 PRs), with a median duration of response (DOR) of 6.1 months. Only one patient received ibrutinib monotherapy in the frontline (This patient had a 17p chromosomal deletion and heavily pretreated CLL.) and achieved a PR with a sustained response of 10.8 months (the longest DOR of the 3 patients) [61].

This case series was followed by a phase I/II clinical trial evaluating acalabrutinib 200 mg BID in 25 patients with RT (14 of whom had relapsed/refractory RT) [60]. The ORR for the entire cohort was 40% (8% CR, 32% PR) with a median duration of response of 6 months (range 0.3–63 months). Interestingly, acalabrutinib in the relapsed/refractory setting seems to have the same efficacy as in the front-line (ORR of 43% versus 36% respectively). While patients who received ibrutinib pretreatment (including at the time of RT) did worse, they still had some response: Ibrutinib pretreated patients had an ORR of 25% with a median DOR of 1.8 months, compared with an ORR of 58% and median DOR of 9.2 months for those without ibrutinib pretreatment. Though acalabrutinib was given at twice the standard dose used for CLL, common BTKi adverse events were not readily seen. Atrial fibrillation was present in only 8% of the population, and all hemorrhagic events were grade 1–2. More problematic were infectious complications: neutropenia occurred in 28% of patients; febrile neutropenia occurred in 12% of patients; and sepsis occurred in 8% of patients. An additional 8% of patients (n = 3) experienced grade 4–5 toxicities related to infection.

The recently published phase I/II BRUIN study investigated the noncovalent BTKi, pirtobrutinib as monotherapy for R/R RT. (In the late stages of the study, the study was expanded to include 10 patients who received pirtobrutinib as frontline treatment for their RT.) Of the 57 patients enrolled in the trial, 50 were evaluable. Of the evaluable patients, the ORR was 54% (5 CRs and 22 PRs) with a duration of response of 8.6 months (though this number is challenging to interpret since median follow-up was 5.5 months and 64% of patients were censored at the time of last follow up) [58].

Altogether, these three studies suggest BTKis have efficacy in RT (even as salvage therapy) with an impressive median survival of 6 months and a good side effect profile. Given these findings, trials involving BTKis in combination with other therapies for RT are underway. There are currently seven ongoing trials evaluating the efficacy of BTKis in combination with other therapies (Table 3).

## 11. Inhibitors of Program Cell Death Receptor-1 (PD-1) and Its Ligand (PD-L1)

Immune checkpoint blockade, whereby monoclonal antibodies can be used to prevent T-cells from recognizing a tumor as self (thus facilitating an immune-mediated attack of the tumor), is a promising therapy for many cancers. The most targeted pathway for disruption involves PD-1 (which is usually expressed on T-cells) and its ligand PD-L1 (which is usually expressed on tumor cells). Though PD-1 is primarily thought of as a self-recognition pathway, as will be described below, there may be more roles for PD-1 and its ligand. Application of checkpoint blockade was first suggested in CLL and RT after researchers discovered populations of immunosuppressed T-cells in the lymph nodes of CLL patients [62]. In 2017, the first trial of checkpoint inhibition for RT was conducted, where nine patients were treated with pembrolizumab monotherapy [63]. Pembrolizumab appeared to have activity on RT-DLBCL cells only, and no activity on the underlying CLL. In turn, some responders developed thrombocytopenia as a result of progressive CLL. Their thrombocytopenia improved with the addition of idelalisib (a PI3K inhibitor), suggesting the importance of combination therapy to treat the underlying CLL. In this study, three interesting observations were made with respect to PD-1/PD-L1 expression: (1) RT-DLBCL cells were more likely to express PD-1 (opposed to PD-L1, which was felt to be expressed by histiocytes in the tumor microenvironment), (2) PD-1/PD-L1 expression was associated with better responses to checkpoint blockade, (3) PD-1/PD-L1 expression was associated with ibrutinib pre-treatment. The fact that PD-1 was found on the surface of the malignant B-cells and PD-L1 was found on the surface of tumor stromal elements suggests there may be alternative roles for PD-1 functionality aside from its role in immunosuppression (Figure 1). Indeed, there have been data that hypothesize PD-L1 expression on stromal elements in tumor microenvironments feeds PD-1 expressing neoplastic cells in other malignancies [64].

A larger phase II study investigating pembrolizumab was published with twice the number of participants (n = 23). Unfortunately, this follow-up study was not able to replicate a 44% ORR, and instead reported a mere 13% ORR. It is difficult to assess the discordant results between these two studies since the latter study did not report prognostic RT variables, nor did they report PD-1 expression (which was shown to be an important determinant in outcome) [65].

To improve upon single agent responses, investigations of checkpoint blockade in combination with other drugs are ongoing. A recently reported clinical trial evaluating the combination of nivolumab with ibrutinib demonstrated a 42% ORR with a median OS of 13 months (n = 24) and very few toxicities [66]. Another investigational strategy involving the combination of pembrolizumab, umbralisib (a PI3Ki) and ublituximab (a type I CD20 antibody) has shown promising initial results with an ORR of 50% in 4 relapsed/refractory RT patients with ongoing remissions of greater than 7 months [67]. A longer follow-up of this study is currently pending (NCT02535286). Additional studies investigating combinations with checkpoint inhibitors are ongoing (Table 3).

## 12. Venetoclax

B-cell lymphoma-2 (BCL-2) is upregulated nearly ubiquitously in CLL and RT. Small molecular mimetics of the BH3 binding domain of BCL-2 were developed and venetoclax approved for CLL in 2016 (Figure 1). The first phase I study evaluating venetoclax monotherapy in relapsed/refractory RT was published in 2017, demonstrating an ORR of 43% (with 0% CR) in 7 patients [68]. A phase II study evaluating venetoclax in combination with dose adjusted R-EPOCH was even more promising with ORR of 61.5% (50% CR) and a median OS of 19.6 months in 25 patients [69]. A joint effort with MDACC and Mayo Clinic has led to further validation that venetoclax has synergistic properties with historic regimens used in RT. In 55 patients evaluated with RT, 10 received venetoclax in combination with R-CHOP (ORR of 60%, CR of 50%); 20 received venetoclax in combination with chemoimmunotherapy (ORR of 50%, CR of 40%); 20 received venetoclax in combination with a BTKi +/− and anti-CD20 antibody (ORR of 40%, CR of 30%); 3 received venetoclax in combination with “varied-based regimens” (ORR/CR not reported); and 2 received venetoclax monotherapy (ORR/CR not reported) [70]. Given the interest of obtaining a CR to proceed with SCT, there is a great desire for venetoclax-based regimens and their high CR rates. Since CR rates for combinations involving R-CHOP and DA-REPOCH seem to be similar, the less toxic R-CHOP regimen may be preferred.

## 13. CAR-T Cell Therapy

The use of activated autologous T-cells engineered to express unique chimeric antigen receptors specific to malignant cells (so called autologous CAR-T cells) has gained a great deal of traction in malignant hematology (Figure 1). There are now even allogeneic CAR-T cells, which bypass the needed to harvest and engineer T-cells from the patient, expediting the administration of therapy. Given their success in the treatment of relapsed/refractory DLBCL, improving the median OS by more than 100% with an ORR of >50% [71,72,73], CAR-T has moved into trials for RT. Though there have only been small phase I single arm studies, a response lasting at least 6mo seems to be observed in >50% of patients who relapsed on multiple lines of therapy for their RT [74,75,76].

## 14. Bispecific Antibodies

An alternative to CAR-T cell therapy is bispecific t-cell engagers (BiTEs). BiTEs are bispecific antibodies that target epitopes on both T-cells and tumors, thereby drawing T-cells into proximity of the tumor to facilitate an immune mediated attack. A BiTE of interest in RT is epcoritamab (a CD3xCD20 antibody), which enables CD3+ T-cells to attack CD20+ RT cells. The ongoing phase Ib/II EPCORE CLL-1 trial (NCT04623541) recently published preliminary results on 10 patients that looks promising. Of the 10 patients treated on this trial, 6 received epcoritamab in the front line and 4 in the refractory setting. The ORR was 60%, and the CR rate was 50%. The follow-up has not been long (≥12 weeks, median follow-up not reported), so it remains to be seen how long these remissions will be sustained. Prognostic information was not reported, and subgroups were not analyzed based on whether or not patients received epcoritamab in the front line [77]. A different study evaluating the CD3xCD19 bi-specific blinatumomab in a small patient population (n = 9) suggested blinatumomab may work best in patients with low PD-1 expression [78]. Blinatumomab was also investigated as an adjective therapy in patients with persistent disease after 2 cycles of R-CHOP: Though difficult to interpret, blinatumomab may have improved outcomes in 44% of these patients [79].

BiTEs represent one application of bispecific antibody technology (Figure 1). Another approach under investigation, which does not directly target T-cells, involves targeting two antigens on the neoplastic cell (making the antibody more specific for the neoplastic cell). Currently, there is an ongoing study looking at TG-1801, a bispecific antibody that targets both CD19 and CD47 on RT cells to enable opsonization and destruction by the immune system (Table 3).

## 15. Other Therapeutic Approaches:

In the early 2000s, radioimmunotherapy was an investigational strategy of interest for b-cell lymphomas [80,81,82]. This approach involves linking a radioactive substance (such as a beta emitter) to an antibody in order to deliver localized radiation to neoplastic cells and their surrounding microenvironment. ^90^Y ibritumomab tiuxetan (a beta emitter linked to an anti-CD20 monoclonal antibody) was the first and only radioimmunotherapeutic agent studied in RT [82]. Unfortunately, this approach delivered unimpressive results with an ORR of 0% in 7 patients.

Monomethyl auristatin E (MMAE) is a potent anti-tubulin agent that blocks mitosis of neoplastic cells. This compound can be attached to a tumor-specific monoclonal antibody in order to avoid off-target toxicity. Such approach is being trialed in RT with zilovertamab vedotin (an anti-receptor tyrosine kinase-like orphan receptor 1 + MMAE conjugate) and polatuzumab vedotin (an anti-CD79 + MMAE conjugate). Finally, an investigational approach assessing the inhibition of the important cell cycle regulator cyclin-D kinase 9 is underway (Table 3).

## 16. Clinical Case Continued

The patient was enrolled in the phase I/II pembrolizumab/ublituximab/umbralisib trial. After five cycles of treatment, he developed significant transaminitis and was switched to umbralisib monotherapy. He completed an additional seven months of umbralisib until ultimately progressing. Subsequently, he was approved for CAR-T cell therapy. He was given ibrutinib for T-cell harvest and was conditioned with polatuzumab-vedotin plus rituximab. After 11 months, the patient developed progressive disease and opted for hospice. In total, this patient with heavily pre-treated clonally related DLBCL, survived two years without any combination chemotherapy or SCT. Pre-treatment with a BTKi may have explained his SUV_max_ < 10 at the time of RT, and his excellent response to an immune checkpoint-based regimen.

## 17. Discussion

Transformation from CLL to a clonally related DLBCL carries a poor prognosis with a median OS of <1 year using historic combination chemotherapy regimens. Newly diagnosed patients with RT-DLBCL may benefit from a clinical trial in the front-line setting since there is currently no best standard of care defined. Though randomized controlled trials are lacking, the best data to-date suggests that using venetoclax in combination with R-CHOP may be the best way to achieve a CR, followed by a consolidative allogeneic stem cell transplant. This strategy may change with ongoing studies. For patients in the relapsed refractory setting, there are several therapeutic options to potentially extend survival six months or more. Evaluation for PD-1/PD-L1 expression may guide practitioners towards or away from PD-1/PD-L1 antibody-based therapies. We report a case in which a patient with heavily pre-treated CLL lived two years after RT diagnosis with the use of targeted and cellular therapy. As our treatments evolve in this modern era, there is hope that survival will continue to improve.

## 18. Conclusions

Despite major advancements over the last 80 years, RT remains a challenging disease to treat. It is difficult to know how much improvement we have made in the overall survival for patients with RT, as studies are ongoing; however, there is great optimism that continued discoveries in the molecular biology of this disease, as well as advancements in targeted therapies and immunotherapy, will lead to improved outcomes for all patients.

## Figures and Tables

**Figure 1 cancers-15-01857-f001:**
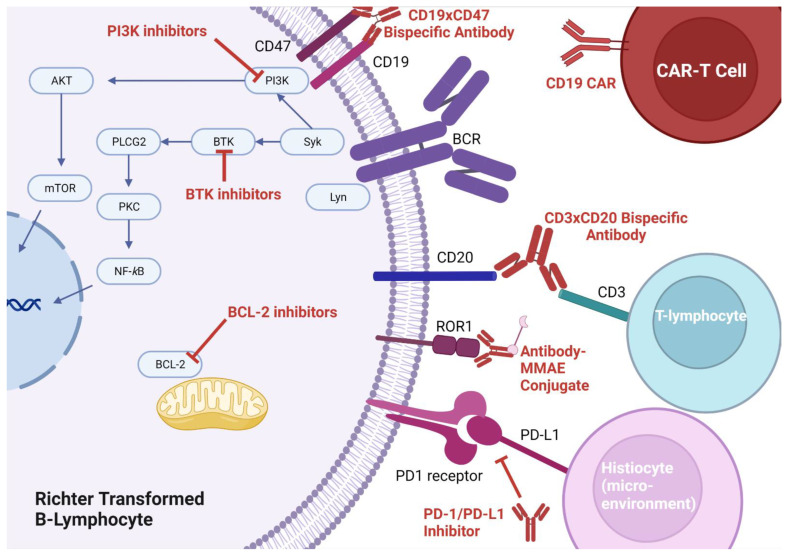
Mechanisms of action for novel targeted therapies in Richter Transformation. BTK inhibitors disrupt signaling from the surface B-cell receptor (BCR) to the canonical nuclear factor κB (NF-κB) pathway. PI3k inhibitors primarily disrupt signaling from surface CD19 to the mammalian target of rapamycin (mTOR) pathway although there is debate that PI3K may interact with other pathways. BCL-2 inhibitors disrupt mitochondrial associated anti-apoptotic pathways and induce cell death. Antibodies directed at program cell death-1 receptor or its ligand (PD-1 and PD-L1 respectively) seem to interfere with tumor microenvironment stimulation of neoplastic cell growth. Bi-specific T-cell engagers (BiTEs) are a class of bispecific antibodies which bring T-cells in proximity of cancer cells (the example shown here is an anti-CD3xCD20 bispecific). Anti-CD19xCD47 antibodies represent another class of bispecific antibodies which target two antigens on a neoplastic cell, making these antibodies more selective for neoplastic cells than traditional monoclonal antibodies. CAR-T cells are engineered T-cells with chimeric receptors designed to target cancer cells (the example shown here is an anti-CD19 CAR T-cell). Antibody-MMAE conjugates allow for the delivery of the potent anti-tubulin drug MMAE, which halts cytoskeletal elements and inhibits cell division.

**Table 1 cancers-15-01857-t001:** Risk factors associated with Richter Transformation in untreated CLL patients.

Authors	CLL Variables	CLL Sample Size (n)	No. of pts w/RT	Median f/u	HR	CI	*p*-Value
Rossi et al. [6]		(185)	(17)	3.9 years			
	Lymph node ≥ 3 cm	26/185	9/17	NR	6.5	(2.3–18.7)	0.001
	No del(13q)	78/173 *	NR	NR	4.1	(1.1–14.7)	0.03
Rossi et al. [20]		(605)	(40)	6.5 years			
	IGHV4-39 usage	NR	NR	NR	3.1	(1.0–9.7)	0.047
	+12/−11q/TP53 disruption	NR	NR	NR	3.3	(1.6–8.7)	0.002
	NOTCH1 mutated	73/605	NR	NR	3.7	(1.6–8.7)	0.002
Visentin et al. [21]		(540)	(28)	6.7 years			
	Binet B-C	133/540	13/28	NR	2.9	(1.4–6.3)	0.004
	Unmutated IGHV	309/540	22/28	NR	4.5	(1.8–11.3)	0.001
	Del(11q)	52/540	6/28	NR	2.8	(1.1–6.9)	0.0285
	TP53 disruption	58/540	9/28	NR	3.9	(1.8–8.7)	0.0008
	Complex Karyotype	107/540	14/28	NR	4.7	(2.2–9.9)	<0.0001

***** Not all 185 patients were evaluable with a CLL FISH. NR, Not reported; RT, Richter Transformation; HR, Hazard ratio; CI, confidence interval 95%.

**Table 2 cancers-15-01857-t002:** Combination chemotherapy regimens.

Regimen	Years of Recruitment	n	Median Age in Years	ORR (%)	CR (%)	Median OS (Months)
Anthracycline-containing regimens						
R-CHOP [39]	2003–2008	15	69 (N/A)	67%	7%	21 months
O-CHOP [40]	2011–2014	37	66 (43–90)	47%	27%	11.4 months
HyperCVXD [41]	NR (published in 2000)	29	61 (36–75)	41%	38%	10 months
Rituximab and GM-CSF with alternating hyperCVAD and MTX/cytarabine [42]	1999–2001	30	59 (27–79)	43%	18%	8.5 months
R-EPOCH [43]	2006–2014	46	67 (38–83)	39%	N/A	5.9 months
Platinum-containing regimens						
OFAR1 [44]	2004–2006	20	59 (34–77)	50%	20%	8 months
OFAR2 [45]	2007–2010	35	63 (40–81)	43%	8.6%	6.6 months
Fludarabine-containing regimens						
PFA or CFA [46]	1992–1996	12	59 (49–74)	45%	N/A	17 months
FACPGM [47]	1997–2001	15	62 (42–74)	5%	5%	2.2 months

CFA, cyclophosphamide–fludarabine–arabinosyl cytosine; FACPGM, fludarabine–cytarabine–cyclophosphamide–cisplatin–GM-CSF; GM-CSF, granulocyte–macrophage colony-stimulating factor; HyperCVAD, fractionated cyclophosphamide–vincristine–liposomal daunorubicin–dexamethasone; MTX, methotrexate; N/A, not available; OFAR, oxaliplatin–fludarabine–cytarabine–rituximab; PFA, cisplatin, fludarabine, cytarabine.

**Table 3 cancers-15-01857-t003:** Ongoing clinical trials in RT.

**BTKi Combinations**	
R-CHOP +/− Acalabrutinib	NCT03899337
Obinutuzumab + Venetoclax + Ibrutinib	NCT04939363
Zanubrutinib + Tislelizumab	NCT04271956
Acalabrutinib + Venetoclax + Durvalumab	NCT05388006
R-EPOCH + Ibrutinib	NCT04992377
Ipilimumab + Ibrutinib + Nivolumab	NCT04781855
Pirtobrutinib + Venetoclax + Obinutuzumab	NCT05536349
**PD-1 inhibitors**	
Pembrolizumab + Ublituximab + Umbralisib	NCT02535286
Zanubrutinib + Tislelizumab	NCT04271956
Acalabrutinib + Venetoclax + Durvalumab	NCT05388006
Ipilimumab + Ibrutinib + Nivolumab	NCT04781855
Duvelisib + Nivolumab	NCT03892044
Copanlisib + Nivolumab	NCT03884998
**PD-L1 inhibitors**	
Atezolizumab + Obinutuzumab + Venetoclax	NCT02846623
Atezolizumab + Obinutuzumab + Venetoclax in RT only	NCT04082897
Atezolizumab + Rituximab + Gemcitabine + Oxaliplatin	NCT03321643
**BCL-2 inhibitor combinations**	
Venetoclax + DA-R-EPOCH or R-CHOP	NCT03054896
Obinutuzumab + Venetoclax + Ibrutinib	NCT04939363
Acalabrutinib + Venetoclax + Durvalumab	NCT05388006
Pirtobrutinib + Venetoclax + Obinutuzumab	NCT05536349
Acalabrutinib + Venetoclax + Durvalumab	NCT05388006
Atezolizumab + Obinutuzumab + Venetoclax	NCT02846623
Atezolizumab + Obinutuzumab + Venetoclax in RT only	NCT04082897
Duvelisib + Venetoclax	NCT03534323
LP-118 *	NCT04771572
**Bispecific antibodies**	
TG-1801 ** +/− Ublituximab	NCT04806035
XmAb13676 ***	NCT02924402
Blinatumumab after R-CHOP	NCT03931642
Epcoritamab	NCT04623541
**Antibody-drug conjugates**	
Zilovertamab Vedotin (MK-2140) +/− Nemtabrutinib	NCT05458297
AutoSCT followed by Polatuzumab Vedotin	NCT04491370
**Cyclin-Dependent Kinase 9 inhibitor**	
VIP152	NCT04978779

BTKi, Bruton’s tyrosine kinase inhibitor; PD-1, programmed death-1; PDL-1, programmed death ligand-1; BCL-2, B-cell lymphoma-2. * LP-118 = BCL2/BCLX inhibitor. ** TG-1801 = anti-CD47/CD19 bispecific antibody. *** XmAb13676 = anti-CD20/CD3 bispecific antibody.

## Data Availability

Not Applicable.

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
