# Peer review of "Treatment of Richter Transformation of Chronic Lymphocytic Leukemia in the Modern Era"

_cancers, 2023, doi:10.3390/cancers15061857_

Round 1
Reviewer 1 Report
The authors are giving a detailed review on Richter's syndrome. The review is well balanced, and cites most of the trials regarding this field.
There is only one minor comment. Correctly the authors are focusing on allogeneic transplantation, and only comment on autologous transplantation. A few more sentences should be included with some relevant literature, to define the role of autologous transplantation, and why is it not the optimal treatment option.
Author Response
We thank the reviewer for this comment. We have now added the below text on Lines 534-539.
In general, alloSCT has been the preferred SCT technique in RT given the precieved graft versus leukemia benefit (whereby the donor’s immune system will attack any residual host cancer cells). It is important to note that some patients may lack a good donor for an alloSCT, or there may be other clinical reasons in which an alloSCT may not be the best approach for a patient. In such circumstances an autoSCT may be considered.
Reviewer 2 Report
The paper by Briski and Taylor is is a clearly written and comprehensive review on the the current knowledge on Richter transformation, from diagnosis to treatment. The literature review is extensive and the presentation well structured and useful for the readers.
I have some general comments:
- It would make sense to have the clinical case in the beginning of the paper, to be able to follow the author's comments on diagnosis and treatment with that example; it may be a bit lost at the end.
-Figure 1 nicely summarises the physiological basis for new treatment strategies; it could include the CD79b and ROR1 antigens, since ADC with those targets are mentioned in the new treatments section
- Table 3, page 8 is a good summary, but some of the trials are repeated under different headings. The drug class of LP118, TG-1801, VIP152 and XmAb13676 should be stated in the legend to facilitate understanding by the reader.
Specific comments are:
- In the Diagnosis section (page 3) a comment should be made on the distinction between Richter and histologically aggressive (or advanced) CLL, also from the clinical point of view. This may have therapeutic implications.
- Also, phenotypic changes that may occur when CLL transforms to Richter were not mentioned. As well, the histology of Richter could be described.
- Regarding RT pathogenesis (lines 110-111), although the series are small and with limited statistical power, the current knowledge on genetic drivers of transformation is progressing and that should be acknowledge.
- In the Prognosis section (page 4), it should be made clear that flow cytometry cannot define clonality, only suggest it. Besides rtPCR, NGS is another useful method to study clonality. The type of samples used in these studies are not mentioned; it is true that isolation of CLL and DLBCL in samples may be difficult, but using peripheral blood may be of help with CLL, and microdissection with DLBCL
- There is a mistake on line 152 (page 4) - the referred study includes 50 clonally related and 13 unrelated patients (the opposite is said in the text)
- the impact of prior therapy with novel agents (BTKi, BCL2i) on the outcomes of RT could be more deeply discussed since double exposed/refractory patients are becoming more frequent.
- Regarding treatment, on the chemotherapy section it needs to be clarified that the current recommended regimen includes rituximab and anthracyclines (RCHOP or similar); other regimens are apparently either less effective or more toxic.
- The limitations of the studies including allogeneic transplantation are highlighted. However, it should be stated hat this is an option available to a strict minority of patients, in an elderly and comorbid population. Also, the impact of MRD+ is unkown (line 245), as explained, but the importance of CR, although only suggested because studies are small, agrees with what is known in other hematological malignancies.
- Could the authors check the references for autologous transplant (line 249)? they may not be correct
- It could be of interest to include in the section of BTK inhibitors the updated results from the BRUIN phase 1/2 study recently presented at ASH 2022 (abstract 347 from Dr. Shah, with 57 RT patients)
- Toxicity of venetoclax combined with DA EPOCH R is considerable and could be commented (page 10, line 355)
Round 2
Reviewer 2 Report
The authors reviewed the paper according to the suggestions. The current versions adequately answers the previous questions.